# Assessing the Turning Ability during Walking in People with Stroke Using L Test

**DOI:** 10.3390/ijerph20043618

**Published:** 2023-02-17

**Authors:** Shamay S. M. Ng, Mimi M. Y. Tse, Peiming Chen, Tony P. S. Lam, Tony H. T. Yeung, Tai-Wa Liu, Billy C. L. So

**Affiliations:** 1Department of Rehabilitation Sciences, The Hong Kong Polytechnic University, Hung Hom, Hong Kong SAR, China; 2Research Centre for Chinese Medicine Innovation, The Hong Kong Polytechnic University, Hung Hom, Hong Kong SAR, China; 3School of Nursing and Health Studies, Hong Kong Metropolitan University, Ho Man Tin, Hong Kong SAR, China

**Keywords:** stroke, lower limb, assessment

## Abstract

Background: The L Test of Functional Mobility (L Test) was developed to assess the advanced mobility, which includes both turning and walking ability. This study aimed to evaluate (1) the intra-rater reliability of the L Test in four turning conditions, (2) the correlation with other stroke-specific impairment for community-dwelling older adults with stroke, and (3) the optimal cut-off completion time of the L Test to distinguish the difference of performance between healthy older adults and people with stroke. Methods: This is a cross-sectional design. Thirty older adults with stroke and healthy older adults were included. The subjects were assessed by L Test along with other stroke-specific outcomes. Results: The L Test showed excellent intra-rater reliability (ICC = 0.945–0.978) for the four turning conditions. There were significant correlations between L Test completion times and Fugl–Meyer Assessment–Lower Extremity (FMA-LE) scores, Fugl–Meyer Assessment–Upper Extremity (FMA-UE) scores, Berg Balance Scale (BBS) score, and Timed Up and Go (TUG) Test scores. The cut-off of the L Test was established as 23.41–24.13 s. Conclusion: The L Test is an easy-to-administer clinical test for assessing the turning ability of people with stroke.

## 1. Introduction

Turning is a basic movement for many everyday activities, such as changing direction during walking to avoid obstacles or navigating crowded environments. Due to the motor impairment led by stroke, people with stroke often show difficulties in turning because of impaired temporal and spatial coordination among head, trunk, and pelvis [1]. The turning ability of stroke patients reveals their ability to reintegrate into the community and live safely at home [2]. Therefore, an effective assessment tool to evaluate the turning ability is crucial in stroke rehabilitation.

Several outcome measures that are currently used in clinical environments to assess the turning abilities include the Mini Balance Evaluation Systems Test (Mini-BESTest), the Berg Balance Scale (BBS), and the Timed Up and Go (TUG) Test. For example, two items of the Mini BESTest—Item 12 (walk with pivot turns) [3] and Item 14 (TUG dual-task test) [4]—measure patients’ turning abilities under various conditions. However, although these tests measure the physical state of patients with high reliability and validity, they all have a major limitation: they measure only the general turning ability of patients, without determining their ability to turn to each side. 

The L Test of Functional Mobility (the L Test) [5] was developed on the basis of the TUG Test. The L Test is named for the fact that the walking path in the test is L-shaped [5]. The L Test incorporates the transfer skill sets used in the TUG Test, as these aid the analysis of patients’ walking abilities and includes a turning motion task that requires patients to turn and walk along a hallway. The L Test incorporates two transfers and four turns, at least one of which must be towards the opposite side. The single condition L Test (stand up from the chair, walk 3 m forward, turn 90 degrees, walk 7 m forward, turn 180 degrees, then walk back along the L-shaped path and sit down on the chair) showed excellent inter-rater reliability and intra-rater reliability (intra-class correlation coefficient (ICC) = 0.990) and significantly correlated with the completion time of the L Test and those of the TUG Test (r = 0.89; *p* < 0.001) and 10 m walk test (r = 0.88; *p* < 0.001) in people with chronic stroke [6].

However, the psychometric property of the L Test in different turning conditions, including turning direction in 90 or 180 degrees, has not been investigated. Moreover, the correlation between the performance in L Test and stroke-specific impairments, and the cut-off score distinguishing the performance in L Test of people with stroke and healthy older adults have not yet been determined.

Accordingly, the current study aimed to evaluate (1) the intra-rater reliability of the L Test in four turning conditions, (2) the correlation with other stroke-specific impairment for community-dwelling older adults with stroke, and (3) the optimal cut-off completion time of the L Test to distinguish the difference of performance between healthy older adults and people with stroke. 

## 2. Materials and Methods

### 2.1. Ethical Consideration 

This was a cross-sectional study. Clear explanations were given to all participants, who gave their written consent before the process of data collection began. The ethics committee of The Hong Kong Polytechnic University approved the protocol of this study (HSEARS20160202006) in February 2016 and all of the study’s procedures were performed according to the principles of the Declaration of Helsinki. 

### 2.2. Sample Size Calculation 

The L Test has excellent intra-rater reliability (ICC = 0.990) for assessing community-dwelling older adults with stroke in a previous study [6]. In the current study, we assumed that an acceptable ICC value for assessing intra-rater reliability in people with stroke is 0.90. It was determined, using an online calculator [7], that a sample size of at least 22 would be required to achieve 80% power and a significance level of 0.05. 

A previous study found that L Test completion time showed significant correlation with the TUG Test (r = 0.89; *p* < 0.001) and 10 m walk test (r = 0.88; *p* < 0.001) in people with chronic stroke [6]. In this study, we assumed that L Test completion time would show significantly and moderately strongly correlation (ρ = 0.5) with the selected stroke-specific outcome measures. A sample size of 21 subjects would be required to achieve 80% power and a significance level of 0.05. The sample size was estimated using G∗Power software, version 3.1.9.7 (Franz Faul, University of Kiel, Kiel, Germany). To increase the power of the current study for determining the ICC of the L Test, 30 people with stroke were recruited.

### 2.3. Participants 

The participants comprised 30 people with stroke (19 males and 11 females) and 32 healthy older adults (11 males and 21 females), who were recruited from a local self-help group for people with stroke via poster advertisements. 

The inclusion criteria for the people with stroke were (i) aged 50 years or above; (ii) diagnosed with a single stroke for more than 1 year; (iii) unilateral paresis; (iv) able to give written consent; (v) able to perform all assessments independently without aids; (vi) an Abbreviated Mental Test Score of at least 7 [8]; and (vii) a stable general medical condition. The exclusion criterion for the people with stroke was having any neurological disorder or comorbidity other than stroke, such as Parkinson’s disease, uncontrolled diabetes, or any uncontrolled cardiovascular or musculoskeletal conditions that might hinder performance on the L Test. 

The inclusion criteria for the healthy older adults were (i) aged 50 years or above and (ii) community dwelling. The exclusion criterion for the healthy older adults was having cardiovascular, musculoskeletal, or neurological deficits. 

### 2.4. Testing Procedure 

To investigate the intra-rater reliability of the L Test, two practice trials and 3 timed trials were conducted for enhancing the accuracy of measurement of participants’ performance and minimizing learning effects. One rater was pre-trained to conduct the L Test in a standardized manner and also measured the participants’ L Test completion times using a digital stopwatch. 

The protocol is shown in Figure 1. First, the demographic data of all participants were collected, and then the L Test was performed. All people with stroke completed the L Test, the Fugl–Meyer Assessment–Lower Extremity (FMA–LE), the Fugl–Meyer Assessment–Upper Extremity (FMA–UE), the handgrip strength test on the paretic and non-paretic side, the Five Times Sit-to-Stand Test (FTSTST), the BBS test, the TUG Test, the Short-Form Health Survey (SF-36), and the Community Integration Measure (CIM) in a random order, which was established by drawing lots. Subjects were provided at least 2 min of rest between each test and longer rest interval was allowed if necessary. The healthy older adults performed only the L Test, and their completion times were compared with those of the people with stroke to determine an L Test cut-off score that distinguished the performance of people with stroke from healthy older adults. 

### 2.5. Outcome Measures

#### 2.5.1. L Test 

L Test was used to assess the turning ability [5]. When the L Test began, subjects were asked to stand from an armless chair, walk 3 m and turn 90°, walk another 7 m, turn 180°, then walk back to the chair following the original path (i.e., total test distance = 20 m), and return to a seated position at the usual walking speed and sit down (See Figure 2) [5]. Standardized instructions have been developed and were explained to patients undergoing testing. After the patients read the instructions, the L Test was demonstrated once to the patients. Subsequently, the patients performed 2 practice tests, followed by the 3 timed tests.

There are a total of 4 conditions in our L Test, as shown below:

Condition 1: the subject turns towards the non-paretic side for the 90° turn and turns towards the non-paretic side for the 180° turn; 

Condition 2: the subject turns towards the non-paretic side for the 90° turn and turns towards the paretic side for the 180° turn; 

Condition 3: the subject turns towards the paretic side for the 90° turn and turns towards the non-paretic side for the 180° turn; 

Condition 4: the subject turns towards the paretic side for the 90° turn and turns towards the paretic side for the 180° turn. 

As mentioned, the inter-rater reliability and intra-rater reliability for L Test assessments of chronic stroke patients are both 0.990, which are excellent [6].

#### 2.5.2. Fugl–Meyer Assessment 

FMA-UE and FMA-LE was used to assess the motor control of upper limb and lower limb in people with stroke [9], respectively. It is a 3-point ordinal scale with 50 items (upper limb: 33 items; lower limb: 17 items), wherein each item is rated from 0 to 2. The total scores for FAM-UE and FMA-LE performance are 66 and 34, respectively, with a higher score indicating better motor control recovery. The FMA has excellent reliability (ICC = 0.85) for people with stroke [10].

#### 2.5.3. Handgrip Strength Test

Handgrip strength was assessed by the hand-held dynamometer. The handgrip strength test assesses strength of the arm [11]. The patient is seated on a chair with forearm and wrist in a neutral position. The patient then squeezes a hand-held dynamometer as hard as possible for each 3 s trial, and the measurement shown on the dynamometer is then recorded. Total of 2 trials were performed, and the results of the 2 trials were averaged. The stronger the handgrip, the higher the muscle strength of the patient’s hand. The handgrip strength test has excellent accuracy and test–retest reliability (ICC > 0.93) [12].

#### 2.5.4. FTSTST

The FTSTST was used to assess the functional muscle strength of the legs [13]. The test requires a patient to stand up and sit down 5 times from a seated position as quickly as possible. The time taken by the patient to complete the five sit-to-stand movements is recorded, with a shorter completion time regarded as indicating a stronger functional muscle strength. This test has high inter-rater and test–retest reliability (ICC = 0.989–0.999) for assessing patients with chronic stroke [13].

#### 2.5.5. TUG Test 

The TUG Test was used to assess the functional mobility [4]. The subjects were asked to get up from a chair, walk 3 m, turn 180°, and walk back to the chair and sit down [14]. The completion time was recorded by stopwatch. Three trials of TUG test were conducted by each subjects. A shorter completion time indicates better higher mobility. The TUG Test showed excellent reliability (ICC > 0.950) for assessing chronic stroke patients [4].

#### 2.5.6. BBS

The BBS was used to assess the functional balance ability. This consists of 14 activities to assess the functional balance of people with stroke [15]. Patients’ performances in each activity are allocated a score from 0 to 4 by therapists, and their scores for all of the activities are summed to give a total score (highest possible score = 56). A higher score indicates better balance and a lower fall risk. The BBS has a high reliability (ICC > 0.95) for assessing patients with stroke [16].

#### 2.5.7. SF-36

The SF-36 was used to assess the health-related quality of life [17]. It is a questionnaire that comprises 36 items across eight subscales and is used to assess the health-related quality of life, which includes both the physical and mental components [17]. A higher score indicates a better health-related quality of life. The SF-36 has excellent reliability for assessing the health-related quality of life of patients [18].

#### 2.5.8. CIM

The CIM was used to assess the community integration level [19]. It is a 10-item ordinal scale, each item rated from 1 (always disagree) to 5 (always agree), with a total score ranging from 10 to 50 [19]. A high score indicates that a higher level of community integration. The CIM has good reliability (ICC = 0.84) for assessing people with chronic stroke [19].

### 2.6. Statistical Analysis

Statistical Package for the Social Sciences software (version 27; IBM Corporation, Armonk, New York, NY, USA) was used to conduct data analysis. Descriptive statistics were used to summarize the demographic data of the participants. The normality of the data and the homogeneity of the variances were assessed using the Shapiro–Wilk test. An independent *t*-test was used to compare the parametric and nonparametric data of both groups of participants. ICC (3,1) were used to assess intra-rater reliability, as the raters were randomly assigned and generalization of results was allowed. ICC values of less than 0.5, 0.5–0.75, 0.75–0.90, and greater than 0.90 were considered to indicate poor, moderate, good, and excellent reliability, respectively [20]. The confidence level for significance was set as α = 0.05.

The correlations between L Test completion time and their handgrip strengths, FMA–LE, FMA–UE, FTSTST, SF-36, BBS, TUG Test, and CIM scores were calculated by Pearson’s *r* for the parametric data and Spearman’s ρ for the nonparametric data. Partial correlation was used to control the demographic factors of age and gender. *r* values less than 0.25, 0.25–0.49, 0.5–0.75, and greater than 0.75 were considered to indicate no, fair, moderate-to-good, and good-to-excellent correlations, respectively. 

To determine the cut-off L Test completion time to distinguish the mobility performance between people with stroke and healthy older adults, the receiver operating characteristic (ROC) curve was plotted, with the trade-off between sensitivity and 1-minus specificity determined using the Youden index [21]. The area under the curve (AUC) was determined and used to calculate discriminative accuracy. AUCs ≥ 0.9, 0.8 to <0.9, 0.7 to <0.8, 0.5 to <0.7, and ≤0.5 were considered to indicate outstanding, excellent, acceptable, poor, and no discriminative accuracy, respectively.

## 3. Results

### 3.1. Characteristics of Participants 

Thirty people with stroke (19 males and 11 females) and 32 healthy older adults (11 males and 21 females) were recruited for this study and had a mean age of 58.0 ± 5.4 years and 62.8 ± 6.3 years, respectively. The characteristics of these two groups of participants are shown in Table 1. The mean body mass index of the people with stroke was 25.2 ± 2.9 and that of the healthy older adults was 22.3 ± 2.9. The mean post-stroke duration was 7.8 ± 4.8 years. There were significant between-group differences in terms of age, sex, weight, and BMI (*p* < 0.05), but not in height (*p* = 0.051). 

Table 2 shows that mean L Test time of the healthy older adults (see remarks) ranged from 18.4 ± 3.3 s to 18.6 ± 3.5 s under Conditions 1 to 4. Table 2 also shows that the mean L Test time of the people with stroke ranged from 32.1 ± 7.4 s to 32.9 ± 8.0 s under Conditions 1 to 4. This indicates that compared with the healthy older adults, the people with stroke took significantly longer to complete the L Test under Conditions 1 to 4 (*p* < 0.001 in all conditions). However, there was no significant within-group difference among the four conditions. 

### 3.2. Reliability of L Test

Table 3 shows the intra-rater reliability of the L Test for the people with stroke. The ICC 95% confidence interval (CI) values for the four conditions ranged from 0.945 (0.901–0.971) to 0.978 (0.961–0.989), which indicates excellent intra-rater reliability. 

### 3.3. Correlation of L Test with Other Stroke-Specific Outcomes

The correlations of the participants’ L Test completion times with the other stroke-specific impairment measures are shown in Table 4. After controlling the factor of gender and age, our correlational analysis revealed that there were significantly negative correlations between the L Test completion time and their FMA–LE and FMA-UE scores and BBS score (r = −0.438 to −0.552, *p* < 0.05), and significantly positive correlation with the TUG Test completion time (r = 0.724 to 0.761, *p* < 0.05) under all four conditions. In contrast, there was no significant correlation between the L Test completion time and the handgrip strength, the FTSTST score, the SF-36 total score, the SF-36 Physical Component Summary score, the SF-36 Mental Health Component Summary score, or the CIM score under any of the conditions.

The optimal cutoff completion time of L Test was found to be 23.41–24.13 s (sensitivity = 96.7–100%, specificity = 90.6–96.9%, AUC = 0.986–0.987, *p* < 0.001) in the four conditions (See Figure 3, Figure 4, Figure 5 and Figure 6). All of these cut-off points showed excellent ability to distinguish the performance between healthy older adults and people with stroke. 

## 4. Discussion

To the best of our knowledge, this was the first study to investigate the psychometric property of the L Test under various turning conditions (Conditions 1 to 4) for assessing the functional mobility of people with stroke living in the community. The L Test showed excellent intra-rater reliability in assessing people with stroke. In addition, the L Test completion time was significantly and negatively correlated with their FMA–LE and FMA-UE scores and BBS score, and positively correlated with the TUG Test completion time under all four conditions.

### 4.1. Performance of L Test in People with Stroke and Healthy Older Adults 

The L Test completion time of the healthy older adults (18.4 ± 3.3 s to 18.6 ± 3.5 s) was significantly shorter than that of the people with stroke (32.1 ± 7.4 s to 32.9 ± 8.0 s). This is consistent with the finding of a previous study [2], which showed that people with stroke took a longer time than the healthy older adults to complete 360° turns. This was likely because people with stroke have lesions in the motor cortex, which reduces the excitability of motor fibers. This results in a reduction in the neuronal firing rate, which causes atrophy of fast-twitch fibers and, hence, reduces muscle strength. In addition, the upper motor neuron lesions also lead to muscle spasticity of muscles due to inappropriate inhibition [22]. These motor impairments led to the poor performance in both walking and turning. Compared with healthy older adults, people with stroke may involve more proportion of step turning (involving changes in the direction to the opposite side of the stance limb) than the pivot turning (involving changes in the direction towards the stance limb) when performing the turning movement [23], as step turning can provide a wider base of support while changing direction than pivot turning. 

Surprisingly, there was no significant difference in the L Test completion times among the four turning conditions with different turning directions in people with stroke in this study. We expected that the motor impairment of the paretic lower limb may lead to poor functions in turning and result in significant differences among four conditions of the L Test completion time. A previous study [24] showed that extensor synergy causes interruption to the initiation and forward propulsion of hip flexion during the swing phase that may lead to inadequate ground clearance in people with stroke. In the absence of forward rotation of the pelvis and hip flexion, the paretic lower limb is brought forward by the circumduction pattern. These turning patterns in people with stroke may cause the longer turning movement. However, compared with the turning direction, the walking phase may play a more important role in the completion time of the L Test. Eventually, the motor impairment in the paretic limb may not cause adverse influences on the L Test in different conditions. The findings indicated that the design of the L Test can eliminate the variances caused by the turning direction. In addition to the design of the L Test, the learning effect may also play a role in the insignificant difference among the four conditions. In this study, the participants were asked to conduct two practice trials and three timed trials. The sufficient practice may minimize the minor difference in the difficulty level among the four conditions. 

It may indicate that either step turns (performed by stepping) or pivot turns (performed without toes leaving the floor) were able to be performed by our subjects with stroke. In addition, the people with stroke may have adopt compensatory strategies to community living to avoid falling. 

The mean L Test completion time of the people with stroke in previous study [6] was 60 ± 28 s, which is significantly longer and more variable than that of people with stroke in this study. The difference of L Test completion time can be attributed to the difference in the motor impairment level between a previous study [6] and our findings. The participants in the current study were recruited from a local self-help group with high functional mobility levels and, thus, might have used combinations of turning strategies in the L Test that were different from those used by the participants in the previous study [6]. Another study [25] found that the subjects with Parkinson’s disease had a mean L Test completion time of 35.4 ± 11.1 s on the first day of their study and that this decreased to 29.7 ± 15.3 s on the second day of their study. Compared with the L Test performance of the subjects with Parkinson’s [25], people with stroke in our study performed slightly better in the L Test. This may be attributable to the pathological difference between Parkinson’s disease and stroke. Typical features of bradykinesia in Parkinson’s disease are different from stroke-specific impairments, including muscle weakness and spasticity which affect only the paretic side [25]. 

### 4.2. Reliability of the L Test 

Consistent with the result of previous studies [6], the ICCs of the L Test results for the people with stroke in our study ranged from 0.945 to 0.977 under Conditions 1 to 4, respectively, which indicates the excellent intra-rater reliability of the L Test in people with stroke. This could be attributable to the standardized protocol to perform the L Test. Our instructor is well-trained and, so, provided clear instructions to the participants and took accurate measurements. In addition, there was a 3 min rest interval between each test to prevent the participants becoming fatigue. Meanwhile, the participants were allowed to perform the L Test for three timed trials after two practice tests. Therefore, the potential learning effects cannot be excluded from this study. 

### 4.3. Correlation of the L Test and Other Stroke-Specific Impairments

The L Test completion time was moderately correlated with the FMA–UE score under all four conditions (*r* = −0.512–−0.552). The FMA–UE was first developed to measure the motor control in people with stroke [26]. Therefore, this significant correlation is unsurprising, as arm swings play an important role in walking; for example, a previous study [27] found that arm swings in gait patterns play a largely stabilizing role when walking. Thus, people with stroke with low scores in FMA–UE may have low muscle activation and a low range of motion in their arm swing, resulting in difficulty of stabilizing themselves during walking and, so, have low walking and turning speeds when performing the L Test. 

The FMA–LE score was also moderately correlated with the L Test completion time of condition 1 (r = −0.438). Analogous to the FMA–UE, the FMA–LE measures the limb synergy and range of motions of the lower limbs of patients. People with stroke who have a higher score in FMA-LE have better control over their lower limb movements, enabling them to move their lower limbs more quickly and precisely (i.e., with better cadence). This accounts for the strong correlation between the L Test and FMA–LE results. However, the correlation between FMA-LE score and L Test completion time of Conditions 2 to 4 did not reach the significant level (*p* = 0.070–0.109). The insignificant finding may be explained by random error caused by the small sample size in this study (*n* = 30). By increasing the sample size, a significant correlation between FMA-LE and L Test completion times of Conditions 2 to 4 would be expected.

Similarly, there was no significant correlation between the L Test completion time and the handgrip strength under all four conditions. The handgrip strength serves as a general indicator of the fitness of their upper limbs, while the L Test completion time represented the turning and walking ability. They are assessing two different domains of motor functions. Thus, it was reasonable there was no significant correlation between handgrip strength and L Test completion time. 

In addition, there was a moderate correlation between the L Test completion time and the BBS score under all four conditions (*r* ranged from −0.464 to −0.542). The BBS was originally developed to assess the balance and fall risk of older adult populations [16]. Therefore, some items in the BBS, such as Item 1 (sitting to standing), Item 4 (standing to sitting) and Item 11 (turning 360°), are similar to components in the L Test. Therefore, the L Test and the BBS assess similar balancing capacities, which can explain the strong correlation between the results for the L Test and the BBS.

Furthermore, the L Test completion time was strongly correlated with the TUG Test completion time under all four conditions (*r* = 0.755–0.796). The TUG Test is used for clinically assessing functional mobility and fall risk, and its main components include sitting-to-standing and walking and turning. These components are also incorporated into the L Test. Therefore, the L Test completion time showed strong correlation with the TUG Test completion time.

Similarly, the L Test completion time had insignificant correlation with their FTSTST results under all four conditions. This finding can be explained by the different focuses of these two tests. For the FTSTST, it focuses only on the sit-to-stand and stand-to-sit abilities. However, the sit-to-stand and stand-to-sit abilities contribute only to a minor part of the L Test, and its main focus was assessing the turning and walking ability. Thus, there was significant correlation identified between the L test and FTSTST in the present study. Further study in recording the completion time on each task (sit-to-stand, walking, 90 degree turn, and 180 degree turn) may help to understand the contribution of each task to the total completion time. Therefore, the above-mentioned insignificant correlation is unsurprising.

Moreover, the L Test completion time had insignificant correlation with SF-36 scores under all four conditions. SF-36 is a subjective questionnaire that measures the health-related quality of life, thus, assessing the walking and turning abilities of patients is not its main focus. This accounts for the insignificant correlation between L Test and SF-36.

Furthermore, the L Test completion time had insignificant correlation with the CIM total score. The CIM is a subjective questionnaire focused on determining patients’ ability to reintegrate into society, such as their sense of belonging to and knowledge of society. These questions have no relation to the walking and turning capacity of patients, which is the main focus of the L Test. Therefore, it is not surprising that there was an insignificant correlation.

### 4.4. Optimal Cut-Off for L Test Completion Time

The results show that all four conditions distinguished healthy older adults from those with chronic stroke, with the AUCs of the ROC curves ranging from 0.986 to 0.987. The L Test has a probability of 98.6–98.7% to distinguish the performance of people with stroke from healthy older adults, which indicated that the L Test is a sensitive and specific assessment with outstanding discriminatory power of the performance of people with stroke and healthy older adults in the test. 

### 4.5. Limitation of the Study

This study has several limitations. Firstly, a previous study [6] of reliability was used for the estimation of the required sample size. Thus, the number of participants might have been insufficient to investigate the validity of the L Test in relation to other stroke-specific impairments adopted in the current study. We recommend future study to further examine the validity of the L Test using larger sample size and stratifying factors, such as stroke lesion site, age, and gender, to establish the validity of L Test in different groups of people with stroke, or adopting specific inclusion criteria, such as stroke lesion area, to avoid overgeneralization of the validity of L Test. 

Secondly, only the turning ability of people with stroke was assessed using the L Test completion time in the present study. Future study can also assess the performance quality, such as number of steps during the turning tasks, to provide data more scientifically robust and translatable. 

Thirdly, this was a cross-sectional study. We investigated only the intra-rater reliability of the L Test in the present study. Thus, we recommend future studies to investigate the inter-rater reliability, such as reassessing the L Test performance after a few months’ interval, to expand our understandings of the reliability of the L Test in people with stroke.

Fourthly, this study focused on community-dwelling older adults with a single stroke who were aged 50 or above and possessed sufficient mental ability (with AMT ≥ 7). Thus, it has yet to be determined whether the psychometric properties of the L Test are generalizable to common stroke population, such as younger people with stroke, hospitalized patients, recurrent stroke, or impaired cognition or unconscious patients with stroke. Thus, future studies should examine the applicability of the L Test to such population.

Fifthly, variations of stroke lesion site have strong implications on gait capacity. However, this study did not record the lesion site, and its impacts on motor function were not considered. For future validation study, we recommend investigating the relationships between territory of stroke and performance on L Test to provide further evidence of the psychometric properties of the L Test.

Finally, there were gender ratio issues in the present study. The male-to-female ratio was approximately 1:2 in the stroke group and 2:1 in the healthy group, respectively. The implication is that our findings of L Test performance may or may not represent the stroke population, as women have higher risk of stroke and better responses to stroke-related treatment. Moreover, the reverse gender ratio between the stroke group and healthy group may or may not have significant impacts on our cut-off score of the L Test. Gender-related differences in muscle strength [28] and functional task skill [29] have been reported in previous studies. Thus, readers are reminded to interpret our findings with caution with regard to the gender ratio issues.

## 5. Conclusions

Turning is an essential element in daily activities. The present study demonstrated that the L Test is an easily administered clinical measure requiring minimal equipment and personnel inputs to assess the turning ability of people with stroke. Moreover, turning ability is associated with stroke-specific impairments, such as the lesion sites and various central nervous pathways, and has profound impacts on stroke rehabilitation. We recommend future studies to further psychometrically examine the relationships between the L Test and other clinical measures consistent with the International Classification of Functioning, Disability and Health, such as neuroimaging and measures for post-stroke level of activities and participation, to provide a scientific basis for designing turning-ability-related interventions, monitoring stroke prognosis, and improving the quality of life of people with stroke. 

## Figures and Tables

**Figure 1 ijerph-20-03618-f001:**
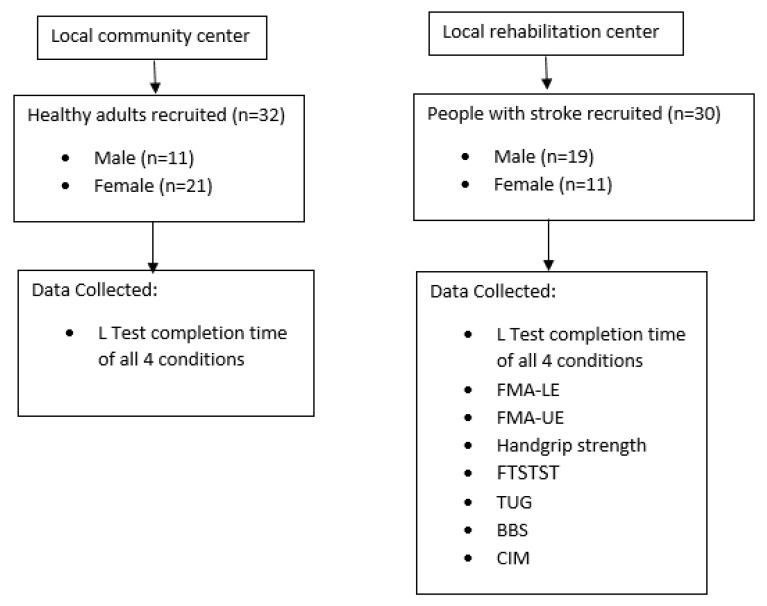
The Assessment Protocol for people with stroke and healthy subjects.

**Figure 2 ijerph-20-03618-f002:**
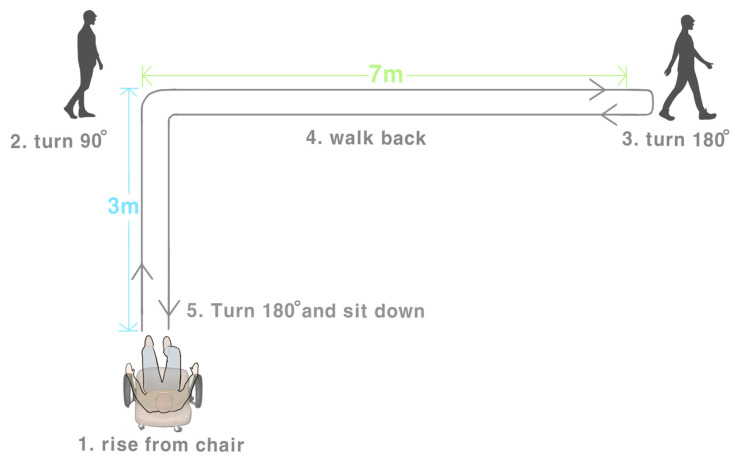
The L Test Protocol.

**Figure 3 ijerph-20-03618-f003:**
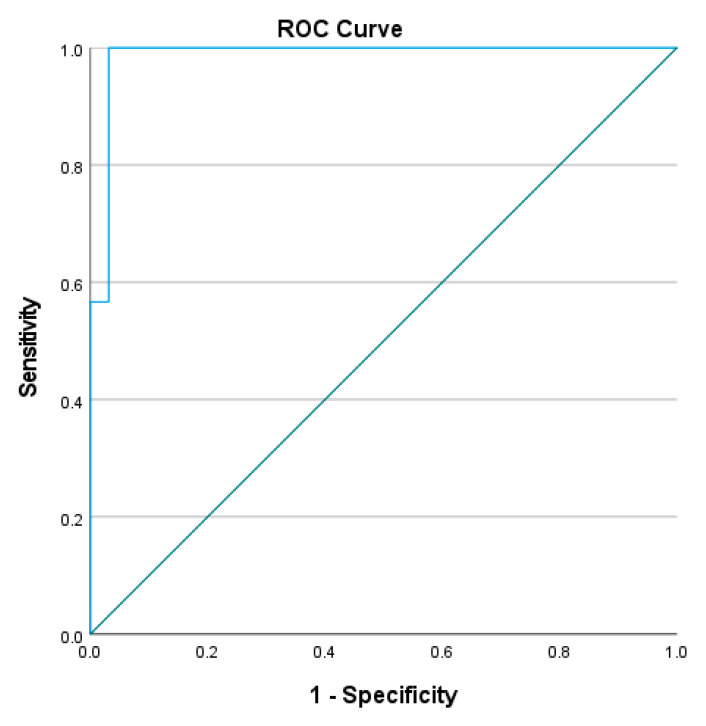
ROC curve of the L Test completion times involving turning towards the non-paretic side for the 90° and then turning towards the non-paretic side for the 180° (AUC = 98.6%, sensitivity = 100%, specificity = 96.9%, *p* < 0.001).

**Figure 4 ijerph-20-03618-f004:**
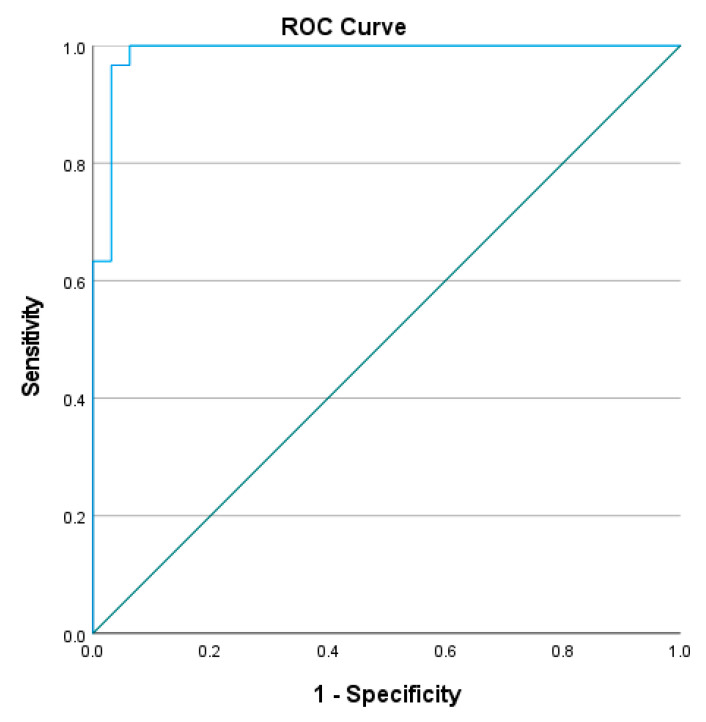
ROC curve of the L Test completion times involving turning towards the non-paretic side for the 90° and then turning towards the paretic side for the 180° (AUC = 98.7%, sensitivity = 96.7%, specificity = 96.9%, *p* < 0.001).

**Figure 5 ijerph-20-03618-f005:**
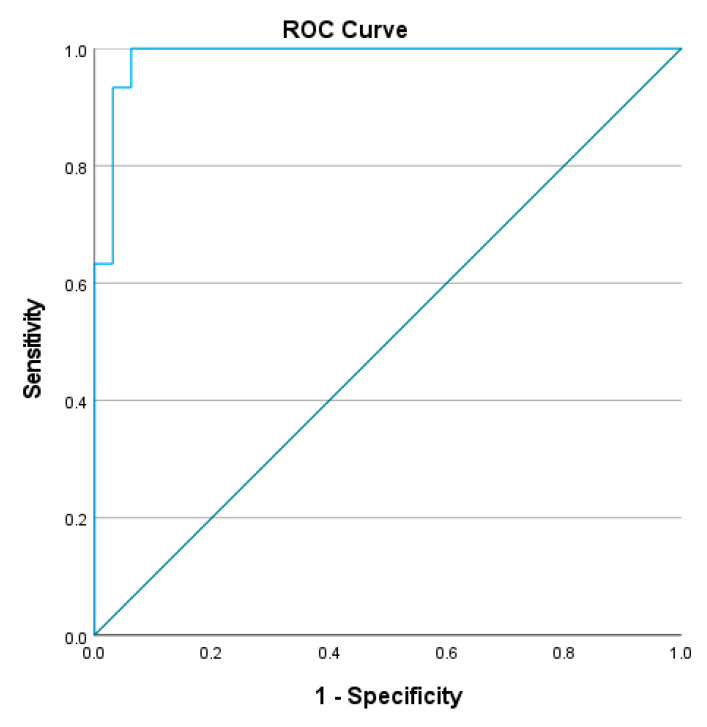
ROC curve of the L Test completion times involving turning towards the paretic side for the 90° and then turning towards the non-paretic side for the 180° (AUC = 98.6%, sensitivity = 100%, specificity = 93.7%, *p* < 0.001).

**Figure 6 ijerph-20-03618-f006:**
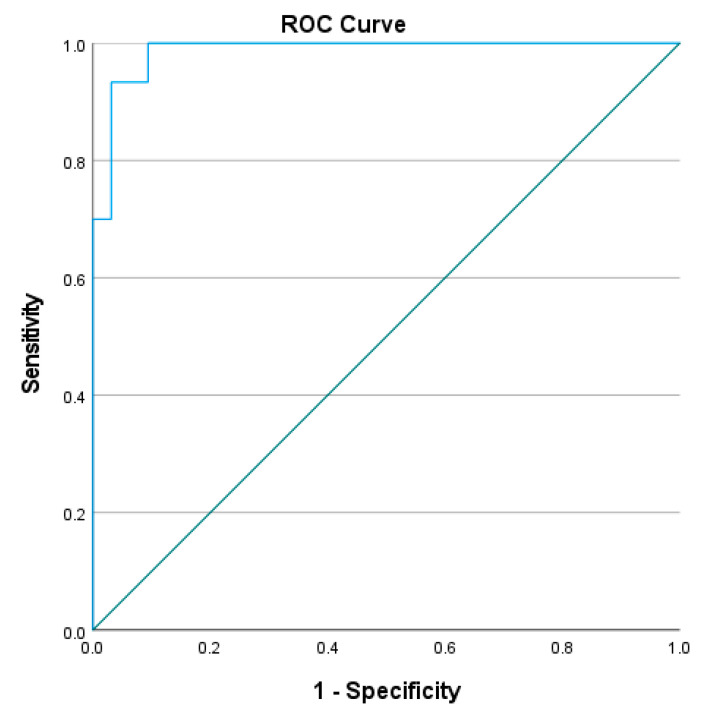
ROC curve of the L Test completion times involving turning towards the paretic side for the 90° and then turning towards the paretic side for the 180° (AUC = 98.6%, sensitivity = 100%, specificity = 90.6%, *p* < 0.001).

**Table 1 ijerph-20-03618-t001:** Demographics of the subjects.

	Stroke (*n* = 30)	Healthy (*n* = 32)	
Age (year, mean ± SD)	58.0 ± 5.4	62.8 ± 6.3	*p* < 0.001
Sex (M/F)	19/11	11/21	*p* = 0.023
Height (cm, mean ± SD)	162.4 ± 8.5	158.9 ± 8.5	*p* = 0.051
Weight (kg, mean ± SD)	66.9 ± 11.4	56.5 ± 9.5	*p* < 0.001
BMI (kg/m^2^)	25.2 ± 2.9	22.3 ± 2.9	*p* < 0.001
Paretic side (L/R)	9/21		
Stroke nature (ischemia/hemorrhage/other)	18/9/3		
Years since stroke (year, mean ± SD)	7.8 ± 4.8		

Remarks: BMI, Body Mass Index; SD, standard deviation; values are expressed as mean ± SD or otherwise noted.

**Table 2 ijerph-20-03618-t002:** Mean scores of L Test in all subjects.

Condition	Healthy (s, Mean ± SD)	Stroke (s, Mean ± SD)	
1	18.4 ± 3.3	32.9 ± 8.0	*p* < 0.001
2	18.4 ± 3.4	32.6 ± 7.6	*p* < 0.001
3	18.6 ± 3.5	32.8 ± 7.4	*p* < 0.001
4	18.5 ± 3.4	32.1 ± 7.4	*p* < 0.001

Remarks: SD, standard deviation; values are expressed as mean ± SD. Condition 1: first 90° turn to left, 180° turn to left for healthy group; both turn to non-paretic side for stroke. Condition 2: first 90° turn to left, 180° turn to right for healthy group; first 90° turn to non-paretic side, 180° turn to paretic side for stroke group. Condition 3: first 90° turn to right, 180° turn to left for healthy group; first 90° turn to paretic side, 180° turn to non-paretic side for stroke group. Condition 4: first 90° turn to right, 180° turn to right for healthy group; both turn to paretic side for stroke.

**Table 3 ijerph-20-03618-t003:** Intra-rater reliability of L Test in people with stroke.

Condition	ICC
1	0.945 (0.901–0.971)
2	0.978 (0.961–0.989)
3	0.965 (0.936–0.982)
4	0.977 (0.959–0.988)

Remarks: ICC, Intraclass Correlation Coefficient; values are ICC (3,1) (95% Confidence Interval).

**Table 4 ijerph-20-03618-t004:** Correlation of L Test Completion Time with other Stroke-Specific Outcome Measures.

Test	Condition 1	*p*	Condition 2	*p*	Condition 3	*p*	Condition 4	*p*
**FMA**								
LE	−0.438 *	0.020	−0.344	0.073	−0.348	0.070	−0.310	0.109
UE	−0.551 *	0.002	−0.534 *	0.003	−0.552 *	0.002	−0.512 *	0.005
**Handgrip**								
Paretic	0.158	0.422	0.122	0.537	0.167	0.396	0.150	0.447
Non-Paretic	0.075	0.703	0.022	0.912	−0.014	0.943	0.024	0.902
FTSTST	0.287	0.139	0.289	0.135	0.213	0.277	0.178	0.364
BBS	−0.512 *	0.005	−0.494 *	0.008	−0.533 *	0.004	−0.488 *	0.008
TUG	0.761 *	<0.001	0.736 *	<0.001	0.735 *	<0.001	0.724 *	<0.001
**SF36**								
Total	−0.038	0.849	−0.064	0.745	0.001	0.995	0.043	0.829
PCS	−0.150	0.445	−0.177	0.369	−0.135	0.495	−0.067	0.733
MCS	0.089	0.651	0.072	0.715	0.136	0.490	0.136	0.491
CIM	−0.356	0.063	−0.350	0.068	−0.302	0.119	−0.255	0.190

Note: BBS, Berg Balance Scale; CIM, Community Integration Measure; FMA, Fugl-Meyer Assessment; FTSTST, Five Time Sit To Stand Test; LE, lower extremity; MCS, mental component; PCS, physical component; SF36, 36-item Short Form Survey; TUG, Timed Up and Go Test; UE, upper extremity; Values are calculated by Spearman’s correlation coefficients (r) unless otherwise specified as r_s_, which is Spearman’s rho. * indicated *p* < 0.05.

## Data Availability

The data are available upon reasonable request to the corresponding author.

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
