# Peer review of "Assessing the Turning Ability during Walking in People with Stroke Using L Test"

_ijerph, 2023, doi:10.3390/ijerph20043618_

Round 1
Reviewer 1 Report
Consider mentioning territory of stroke: middle cerebral artery , posterior circulation, basilar or vertebral or cerebellar
the locations of stroke is important to gait changes
Author Response
Point-to-point response for Reviewer 1
Comment 1: Consider mentioning territory of stroke: middle cerebral artery, posterior circulation, basilar or vertebral or cerebellar the locations of stroke is important to gait changes
Response:
We agreed that stroke lesion site had strong implications on gait changes and added the below paragraph in Page 13-14, line 446 – 450 as below:
“Fifthly, variations of stroke lesion site have strong implications on gait capacity. However, this study did not record the lesion site and its impacts on motor function were not considered. For future validation study, we recommend to investigate the relationships between territory of stroke and performance on L-test to provide further evidence of the psychometric properties of the L test.”
Reviewer 2 Report
This study is testing the use of L-Test for measuring motor impairment in patients with stroke. The study have deigned using a sufficient sample size and inter-observer validity to standardize the test. Results revealed a high inter-rater validity for L-test at 23.4-24s and has proposed as a measure of motor function specifically the turning ability. Overall, the study design and description of tests and results is sufficient, however this study is a form of description and doesn't add a new findings, such as how this test measures the CNS pathways involved in motor function and how using this test can improve quality of care/ treatment, or even if this test could be used as a prognostic tool for motor function rehab. In addition, stroke is a very general terminology. More exclusive criteria to have a specific group of patients or even using a larger sample size with a stratification of patient population according to the localization of injury and age/ gender of patients would have been more helpful. I also would suggest repeating the test after a year/ few months to compare with the original results, which gives a better inter-observer validity. In general, most of my suggestions are not applicable since data has already been collected. I recommend adding some new analysis by removing age and gender as confounding factors.
English writing needs minor revision.
Author Response
Point-to-point response for Reviewer 2
Comment 1: Overall, the study design and description of tests and results is sufficient, however this study is a form of description and doesn't add a new findings, such as how this test measures the CNS pathways involved in motor function and how using this test can improve quality of care/ treatment, or even if this test could be used as a prognostic tool for motor function rehab.
Response: We revised the conclusion session to describe the clinical significance of validating the L Test in Page 14, Line 462-472 as below:
“Turning is an essential element in daily activities. The present study demonstrated that L Test is an easily administered clinical measure requiring minimal equipment and manpower inputs to assess the turning ability of people with stroke. Moreover, turning ability is associated with stroke specific-impairments, such as the lesion sites and various central nervous pathways, and has profound impacts on stroke rehabilitation. We recommend future studies to further psychometrically examine the relationships between L Test and other clinical measures consistent with the International Classification of Functioning, Disability and Health, such as neuroimaging and measures for post-stroke level of activities and participation, to provide scientific basis for designing turning ability related interventions, monitoring stroke prognosis and improving the quality of life of people with stroke.”
Comment 2: In addition, stroke is a very general terminology. More exclusive criteria to have a specific group of patients or even using a larger sample size with a stratification of patient population according to the localization of injury and age/ gender of patients would have been more helpful. I also would suggest repeating the test after a year/ few months to compare with the original results, which gives a better inter-observer validity.
Response: We added these points in our limitation in Page 13-14, Line 424-431 and Line 436 – 439 as below:
Line 424 – 431:
“This study has several limitations. Firstly, a previous study of reliability was used for the estimation of the required sample size. Thus, the number of participants might have been insufficient to investigate the validity of the L Test in relation to other stroke-specific impairment adopted in the current study. We recommend future study to further examine the validity of the L Test using larger sample size and stratifying factors, such as stroke lesion site, age and gender, to establish the validity of L-test in different groups of people with stroke, or adopting specific inclusion criteria, such as stroke lesion area, to avoid overgeneralization of the validity of L-test.”
Line 436 – 439:
“Thirdly, this was a cross-sectional study. We only investigated the intra-rater reliability of the L Test in the present study. Thus, we recommend future studies to investigate the inter-rater reliability, such as re-assess the L-test performance after a few months’ interval, to expand our understandings of the reliability of the L Test in people with stroke.”
Comment 3: In general, most of my suggestions are not applicable since data has already been collected. I recommend adding some new analysis by removing age and gender as confounding factors.
Response: We performed the correlational analysis again by controlling the confounding factors of age and gender. It was described in the statistical analysis in Page 7, Line 206-209 as below:
“The correlation between L Test completion time and their hand grip strengths, FMA–LE, FMA–UE, FTSTST, SF-36, BBS, TUG Test and CIM scores was calculated by Pearson’s r of the parametric data and Spearman’s ρ of the nonparametric data. Partial correlation was used to control the demographic factors of age and gender.”
Reviewer 3 Report
The purpose of the study was to evaluate the intra-rater reliability of the L Test in 4 turning conditions, the correlation with other stroke-specific impairment for older adults with stroke, and the optimal cut-off completion time of the L Test to distinguish the difference of performance between healthy older adults and people with stroke. Considering that the psychometric property of the L Test in different turning conditions was not investigated and the correlation of performance in the L Test with other instruments in stroke-specific impairment and the cut-off time of the L Test which could distinguish patients with stroke from health older adults have yet to be done, this study could be a fresh attempt.
The discussion was based on the thorough analytic strategy, well-organized, and easy to follow. As a reviewer, I would like the authors to consider the following minor points before publishing the manuscript.
1. (line 73) Please include protocol code and the date of approval.
2. (line 210) Please include relevant citation for the paragraph regarding "Youden index" and insert the very reference in the reference section.
3. Table 1. Please revise the measure unit for Weight. (perhaps it should be 'Kg' instead of 'cm'.) And please explain the abbreviations (for I/H) at the bottom of the Table.
4. (line 242) Put parenthesis after (CI) or delete the (CI).
5. (under 3.3, line 247 ~ ) It seems to be helpful for readers to elaborate on the directions of the correlation results. Please cosider revising the paragraph accordingly such as "Their L Test completion was significantly and negatively correlated with their FMA-LL ~ ~ ~, BBS score, and positively correlated with TUG Test completion time under all 4 conditions."
6. Discussion (line 287). The L Test completion time was negatively correlated with FMA-LL, FMA-Ul, paretic side handgrio strength, and BBS score, but positively correlated with TUG Test completion time. Please revise.
7. Conclusion (line 408). Please remove "test-retest reliabilities", which doesn't seem relevant to this study.
8. (line 410). Please replace "their non-paretic side" with "their paretic side".
Author Response
Point-to-point response for Reviewer 3
General Comment: English writing needs minor revision.
Responses: We revised several sentences to enhance the academic writing as suggested by Reviewer 4.
Comment 1: (line 73) Please include protocol code and the date of approval.
Response: We added the ethical approval code (HSEARS20160202006) and date of approval (Feb 2016) in Page 2, Line 72-75 of the manuscript.
“The ethics committee of The Hong Kong Polytechnic University approved the protocol of this study (HSEARS20160202006) in February 2016 and all of the study’s procedures were performed according to the principles of the Declaration of Helsinki.”
Comment 2: (line 210) Please include relevant citation for the paragraph regarding "Youden index" and insert the very reference in the reference section.
Response: We added the below reference for the Youden’s index in Page 7, Line 215.
- Fluss, R.; Faraggi, D.; Reiser, B., Estimation of the Youden Index and its associated cutoff point. Biometrical Journal: Journal of Mathematical Methods in Biosciences 2005, 47, (4), 458-472.
Comment 3: Table 1. Please revise the measure unit for Weight. (perhaps it should be 'Kg' instead of 'cm'.) And please explain the abbreviations (for I/H) at the bottom of the Table.
Response: We revised the measuring unit (kg instead of cm) and added the explanation of abbreviations for I/H in Table 1.
Comment 4: (line 242) Put parenthesis after (CI) or delete the (CI).
Response: We added the parenthesis after (CI) in Page 8, Line 247.
Comment 5: (under 3.3, line 247 ~ ) It seems to be helpful for readers to elaborate on the directions of the correlation results. Please consider revising the paragraph accordingly such as "Their L Test completion was significantly and negatively correlated with their FMA-LL ~ ~ ~, BBS score, and positively correlated with TUG Test completion time under all 4 conditions."
Response: We revised the below sentences in the result part in Page 8, Line 252- 256.
“After controlling the factor of gender and age, our correlational analysis revealed that there were significantly negative correlations between the L Test completion time and their FMA–LE, FMA-UE scores and BBS score (r=-0.438 to -0.552, p<0.05), and significantly positive correlation with the TUG Test completion time (r=0.724 to 0.761, p<0.05) under all 4 conditions.”
Comment 6: Discussion (line 287). The L Test completion time was negatively correlated with FMA-LL, FMA-Ul, paretic side handgrip strength, and BBS score, but positively correlated with TUG Test completion time. Please revise.
Response: We performed the correlational analyses again and revised the discussion part in Page 10, Line 288-291.
“In addition, the L Test completion time was significantly and negatively correlated with their FMA–LE, FMA-UE scores and BBS score, and positively correlated with the TUG Test completion time under all 4 conditions.”
Comment 7: Conclusion (line 408). Please remove "test-retest reliabilities", which doesn't seem relevant to this study.
Response: We revised the conclusion part and the sentence using the words “test-retest reliabilities” was removed as below:
“Turning is an essential element in daily activities. The present study demonstrated that L Test is an easily administered clinical measure requiring minimal equipment and manpower inputs to assess the turning ability of people with stroke. Moreover, turning ability is associated with stroke specific-impairments, such as the lesion sites and various central nervous pathways, and has profound impacts on stroke rehabilitation. We recommend future studies to further psychometrically examine the relationships between L Test and other clinical measures consistent with the International Classification of Functioning, Disability and Health, such as neuroimaging and measures for post-stroke level of activities and participation, to provide scientific basis for designing turning ability related interventions, monitoring stroke prognosis and improving the quality of life of people with stroke.”
Comment 8: (line 410). Please replace "their non-paretic side" with "their paretic side".
Response: As suggested by reviewer 2, we controlled the factor of gender and age when conducted the correlation analysis. We found that there was no significant correlation between L Test and hand grip strength (neither paretic nor non-paretic side). Thus, we revised the findings and deleted the words "their non-paretic side of hand grip strength”.
Reviewer 4 Report
The authors of this manuscript use the psychometric properties of the L Test of functional mobility under multiple turning events to investigate the functional mobility of older stroke survivors living in the community. In contrast to traditional evaluation techniques, the four conditions of the L-test assess the ability of the experimental groups to turn on the paretic and non-paretic sides. The authors set out to define three main goals: intra-rater reliability of the L Test in four different conditions, the relationship between healthy and stroke-impaired older patients living in the community, and the optimal cut-off time for completing the L-test, which is critical for detecting differences. The L-test has good intra-rater reliability, according to the authors. They also report a strong correlation between the BBS score, TUG Test completion time, paretic side handgrip strength, FMA-LL, FMA-UL, and BBS score with the L test completion time. The study is overall well designed, but the following concerns need to be adequately addressed:
The questions addressed in the study are reasonable for using the L-test as a functional mobility test, but the paper has some limitations.
- Several sentences of the introduction and discussion lack clarity. Lines 60-62 is an example of this. These run-on sentences are difficult to understand.
- Line 73: The authors would benefit from mentioning which local institution approved the study protocol.
- Lines 91-93: Did most of the patients suffer one or more strokes? Implications on the L-test should be discussed in the discussion
- Lines 91-93: Did most patients have unilateral paresis?
- Lines 110-116: In the stroke cohort, were all the tests administered together? Is there concern for fatigue?
- Line 144: Inconsistent abbreviation. Previously abbreviated as FMA-LE or FMA-UE only
- Lines 165-171: The TUG test is a commonly used tool for analysis of functional mobility in patients with stroke. Were the four conditions also applied to the TUG test for optimal comparison of the L-test.
- Line 230: The male to female ratio in the healthy and stroke groups are reversed. What are the implications of this, given that women are at a higher risk of stroke and respond differently to treatments? (https://doi.org/10.1161/CIRCRESAHA.121.319915).
- Lines: 45 and 230-231: The authors suggest that the major limitations of the currently administered measures are the ability to measure general turning ability without differentiation of both sides. However, the authors find no significant differences between the four turning conditions. The possible cause for this and the true advantage of the L-test should be further discussed.
- Lines 304-306: According to the authors, elderly stroke patients develop compensatory mechanisms for motor (turning) performance. This should be discussed because it is important to understand compensatory strategies that are unique to older adults when compared to other stroke populations.
- Lines 311-313 and 327-328: Because there is a potential for shorter completion times with each practice (Parkinson's study), there is also a risk of learning and memory loss over the 5 times the test was administered (2 test and 3 timed), which may explain the lack of differences between the four conditions. Discuss
- Lines 370-374: The authors suggest that the lack of correlation between the L-test and FTSTST are the differences in the mobility function measured. However, stand and sit is a component of the L-test. Better justification is necessary here.
- Line 388: The L-test can be used to evaluate mobility function, but it cannot be used as a 'diagnostic' test. The authors should think about rewriting this.
- Line 390: The manuscript only compares the patients' completion times with the available stroke tests, not their performance quality. A better comparison of the type and quality of movement is required to make the data more scientifically robust and translatable.
Author Response
Point-to-point response for Reviewer 4
Comment 1: Several sentences of the introduction and discussion lack clarity. Lines 60-62 is an example of this. These run-on sentences are difficult to understand.
Response: We revised the sentences as below in Page 2, Line 60-63
“Moreover, the correlation between the performance in L Test and stroke-specific impairments, and the cut-off score distinguishing the performance in L Test of people with stroke and healthy older adults have not yet determined.”
Comment 2: Line 73: The authors would benefit from mentioning which local institution approved the study protocol.
Response: We added the local institution “The Hong Kong Polytechnic University” in Page 2, Line 72-75 as below:
“The ethics committee of The Hong Kong Polytechnic University approved the protocol of this study (HSEARS20160202006) on February 2016 and all of the study’s procedures were performed according to the principles of the Declaration of Helsinki.”
Comment 3: Lines 91-93: Did most of the patients suffer one or more strokes? Implications on the L-test should be discussed in the discussion
Response: All subjects included in this study were those who had a single stroke for more than 1 year. We revised the inclusion criteria in Page 3, Line 96 and discussion in Page 13, Line 440-441 as below.
In the participant part:
“(ii) diagnosed with a single stroke for more than 1 year”
In the discussion part:
“Fourthly, this study focused on community-dwelling older adults with a single stroke who were aged 50 or above and possessed sufficient mental ability (with AMT ≥ 7).”
Comment 4: Lines 91-93: Did most patients have unilateral paresis?
Response: All the stroke subjects in the present study were unilateral paresis. This was one of our inclusion criteria and we added the information in Page 3, Line 96 as below:
“(iii) unilateral paresis;”
Comment 5: Lines 110-116: In the stroke cohort, were all the tests administered together? Is there concern for fatigue?
Response: All tests were completed in the same day and at least 2 minutes rest were allowed between tests to minimize the possibility of fatigue. We added more details about how to minimize the fatigue effect in Page 3, Line 118-119.
“Subjects would be provided at least 2 minutes of rest between each test and longer rest interval was allowed if necessary.”
Comment 6: Line 144: Inconsistent abbreviation. Previously abbreviated as FMA-LE or FMA-UE only
Response: We revised the abbreviation in Page 5, Line 148 as below”:
“FMA-UE and FMA-LE was used to assess the motor control of upper limb and lower limb in people with stroke [9], respectively.”
Comment 7: Lines 165-171: The TUG test is a commonly used tool for analysis of functional mobility in patients with stroke. Were the four conditions also applied to the TUG test for optimal comparison of the L-test.
Response: As suggested by the guideline of standardized TUG, we did not require the subjects to conduct the TUG in 4 different turning conditions. We only required the subjects to complete the test at their normal pace.
Comment 8: Line 230: The male to female ratio in the healthy and stroke groups are reversed. What are the implications of this, given that women are at a higher risk of stroke and respond differently to treatments? (https://doi.org/10.1161/CIRCRESAHA.121.319915).
Response:
We added the below sentences to remind the readers to interpret our findings with caution to the gender ratio as below it in the limitation part in Page 14, Line 451-459.
“Finally, there were gender ratio issues in the present study. The male to female ratio was around 1:2 in the stroke group and 2:1 in the healthy group, respectively. The implication is that our findings of L Test performance may or may not represent the stroke population as women have higher risk of stroke and better responses to stroke related treatment. Moreover, the reverse gender ratio between the stroke group and healthy group may or may not have significant impacts on our cut-off score of L Test. Gender-related differences in muscle strength and functional task skill have been reported in previous studies. Thus, readers are reminded to interpret our findings with cautions to the gender ratio issues.”
Comment 9: Lines: 45 and 230-231: The authors suggest that the major limitations of the currently administered measures are the ability to measure general turning ability without differentiation of both sides. However, the authors find no significant differences between the four turning conditions. The possible cause for this and the true advantage of the L-test should be further discussed.
Response: In this study, no significant difference was found between the 4 turning conditions, which may indicated that the 4 conditions of L test may be equivalent to assess the walking and turning ability in people with stroke. We added more details in the discussion in Page 11, Line 309-324 as below:
“We expected that the motor impairment of the paretic lower limb may lead to poor func-tions in turning, and resulted in significant differences among 4 conditions of the L Test completion time. Previous study [24] showed that extensor synergy causes interruption to the initiation and forward propulsion of hip flexion during the swing phase that may lead to inadequate ground clearance in people with stroke. In the absence of forward rotation of the pelvis and hip flexion, the paretic lower limb is brought forward by the circumduction pattern. These turning pattern in people with stroke may cause the longer turning move-ment. However, compared with the turning direction, the walking phase may play a more important role in the completion time of L Test. Eventually, the motor impairment in the paretic limb may not cause adverse influences to the L Test in different conditions. The findings indicated that the design of the L Test can eliminate the variances caused by the turning direction. In addition to the design of the L test, the learning effect may also play a role in the insignificant difference between the 4 conditions. In this study, the participants were asked to conduct 2 practice trial and 3 timed trial. The sufficient practice may help to develop the compensation strategy, which may minimize the minor difference in the dif-ficulty level among the 4 conditions.”
Comment 10: Lines 304-306: According to the authors, elderly stroke patients develop compensatory mechanisms for motor (turning) performance. This should be discussed because it is important to understand compensatory strategies that are unique to older adults when compared to other stroke populations.
Response: We added the explanation as followed: (1) the basic motor impairment led by stroke; (2) how the motor impairment influence the turning strategy in L Test in people with stroke. The change made in Page 11, Line 301-306 was listed below:
“These motor impairments led to the poor performance in both walking and turning. Compared to healthy older adults, people with stroke may involve more proportion of step turning (involving changes in the direction to the opposite side of the stance limb) than the pivot turning (involving changes in the direction towards the stance limb) when performing the turning movement, as step turning can provide a wider base of support while changing direction than pivot turning.”
Comment 11: Lines 311-313 and 327-328: Because there is a potential for shorter completion times with each practice (Parkinson's study), there is also a risk of learning and memory loss over the 5 times the test was administered (2 test and 3 timed), which may explain the lack of differences between the four conditions. Discuss
Response: We added the possible effect of learning effect in the finding among the 4 conditions of L Test accordingly in the manuscript in Page 11, Line 320-324 as below:
“In addition to the design of the L test, the learning effect may also play a role in the insignificant difference between the 4 conditions. In this study, the participants were asked to conduct 2 practice trial and 3 timed trial. The sufficient practice may minimize the minor difference in the difficulty level among the 4 conditions.”
Comment 12: Lines 370-374: The authors suggest that the lack of correlation between the L-test and FTSTST are the differences in the mobility function measured. However, stand and sit is a component of the L-test. Better justification is necessary here.
Response: We revised the sentences below to better justify our findings in Page 12-13, Line 396-401:
“Similarly, the L Test completion time had insignificant correlation with their FTSTST results under all 4 conditions. This finding can be explained by the different focuses of these 2 tests. For the FTSTST, it only focuses on the sit-to-stand and stand-to-sit abilities. However, the sit-to-stand and stand-to-sit abilities only contribute to a minor part of the L Test and the main focus of it was assessing the turning and walking ability. Thus, there was significant correlation identified between the L test and FTSTST in the present study.”
Comment 13: Line 388: The L-test can be used to evaluate mobility function, but it cannot be used as a 'diagnostic' test. The authors should think about rewriting this.
Response: We revised the sentence in Page 13, Line 419-422 as below:
“The L test has a probability of 98.6-98.7% to distinguish the performance of people with stroke from healthy older adults, which indicated that the L test is a sensitive and specific assessment with outstanding discriminatory power of the performance of people with stroke and healthy older adults in the test”.
Comment 14: Line 390: The manuscript only compares the patients' completion times with the available stroke tests, not their performance quality. A better comparison of the type and quality of movement is required to make the data more scientifically robust and translatable.
Response: We added this have revised it in Page 13, Line 432-435.
“Secondly, only the turning ability of people with stroke was assessed using the L test completion time in the present. Future study can also assess the performance quality, such as number of steps during the turning tasks, to provide data more scientifically robust and translatable.”
Round 2
Reviewer 4 Report
The manuscript has been sufficiently revised and will be of interest to readers if published